# NAVIGATING SCALING LAWS: ACCELERATING VISION TRANSFORMER'S TRAINING VIA ADAPTIVE STRATEGIES

## ABSTRACT

In recent years, the state-of-the-art in deep learning has been dominated by very large models that have been pre-trained on vast amounts of data. The paradigm is very simple: Investing more computational resources (optimally) leads to better performance, and even predictably so; neural scaling laws have been derived that accurately forecast the performance of a network for a desired level of compute. This leads to the notion of a "compute-optimal" model, i.e. a model that allocates a given level of compute during training optimally to maximise performance. In this work, we extend the concept of optimality by allowing for an "adaptive" model, i.e. a model that can change its shape during the course of training. By allowing the shape to adapt, we can optimally traverse between the underlying scaling laws, leading to a significant reduction in required compute to reach a given target performance. We focus on vision tasks and the family of Vision Transformers, where the patch size as well as the width naturally serve as adaptive shape parameters. We demonstrate that, guided by scaling laws, we can design compute-optimal adaptive models that beat their "static" counterparts.

## 1 INTRODUCTION

Deep learning has gradually undergone a shift in paradigm, where instead of training specialized models for a given task, a so-called frontier model is fine-tuned. Frontier models are typically defined by their large-scale architectures, often rooted in the Transformer architecture (Vaswani et al., 2017). Their training process involves exposure to extensive and diverse datasets, yielding remarkable advancements in both natural language understanding (OpenAI, 2023; Köpf et al., 2023) and computer vision tasks (Dehghani et al., 2023a; Chen et al., 2023). An inherent and pivotal feature of such models lies in their scalability, whereby their performance can be reliably predicted as a power law across the number of parameters, the volume of data or computational resources utilized (Cortes et al., 1993; Hestness et al., 2017; Rosenfeld et al., 2019; Kaplan et al., 2020). These principles are succinctly encapsulated by the *neural scaling laws* that motivate the choice of a particular model and dataset size given a fixed budget of training compute (Hoffmann et al., 2022).

The ability to accurately predict performance offers an undeniable reassurance in the often uncertain world of deep learning. It nevertheless, introduces an intimidating realization;

*Given a training scheme, a fixed further improvement in performance requires exponentially more compute or parameters.*

While Moore's Law has been a guiding principle in the semiconductor industry for decades, contemporary advancements in machine learning require computational resources surpassing its projections. This disparity highlights the pressing issue of resource allocation, as staying competitive in the realm of deep learning increasingly depends on the availability of substantial computational power. Finding solutions to address this issue becomes increasingly paramount. Delving deeper into the preceding statement, we highlight a pivotal assumption: *the shape of the model*, and therefore the number of FLOPs for a forward-pass remain *fixed throughout the training process*. By "shape" we refer to any characteristic of a model that can be smoothly varied throughout training without leading to strong deterioration in performance (e.g. width, depth or patch size). Such a static ap-

proach (i.e. where model shape remains fixed) may however not always be optimal. For example, it has already been observed that the optimal model size grows smoothly with the loss target and the compute budget (Kaplan et al., 2020).

This paper challenges the assumption of a static model outlined above and explores adaptable training methodologies designed to *surpass conventional scaling laws*. In other words, our aim is to achieve equivalent performance for a specified model with fewer computational resources (FLOPs) than initially projected. To that end, we adapt the shape of the model throughout training, allowing it to optimally traverse between different scaling laws. This enables us to leverage the optimality of all shape configurations in different regions of compute, leading to a more efficient scaling of the model. We train Vision Transformers (Dosovitskiy et al., 2020) and use both model width and patch size as adaptive shape parameters throughout training. We practically showcase how such an adaptive training scheme can lead to substantial training FLOPs reduction, in cases more than $50\%$. In more detail, our contributions are the following:

- We introduce a simple and effective strategy to traverse scaling laws, opting for the one that leads to the faster descent, i.e. maximum performance gain for the same amount of compute.

- We showcase the efficiency of our approach by optimally scheduling the patch size, as well as the width, of a Vision Transformer, leading to significant reductions in the required amount of compute to reach optimal performance.

## 2 RELATED WORK

Neural scaling laws (Cortes et al., 1993), describe how a neural network's performance varies as a power law $E = a(P + d)^b + c$ where $P$ can be either the number of parameters in the model, the number of training samples or simply the number of FLOPs used for training (Rosenfeld et al., 2019). Subsequently, scaling laws have been successfully demonstrated in a range of different applications, including language (Kaplan et al., 2020; Hoffmann et al., 2022) and vision (Zhai et al., 2022; Bachmann et al., 2023), as well as numerous learning settings, including supervised training, generative modelling (Henighan et al., 2020) and transfer learning (Hernandez et al., 2021). The predictive power of scaling laws has also been leveraged to determine compute-optimal models before training; the size of the *Chinchilla* model and the number of training tokens were chosen based on the underlying scaling law and indeed, *Chinchilla* outperformed its larger but sub-optimally trained counterpart *Gopher* (Hoffmann et al., 2022). The training of GPT-4 has also been guided by scaling laws built from training runs of smaller models (OpenAI, 2023).

In this paper, we focus on vision applications and use Vision Transformers (ViTs) as the family of models. Built upon the Transformer architecture used in natural language processing (Vaswani et al., 2017), ViTs have established themselves as the predominant vision architecture for large-scale pretraining tasks (Dehghani et al., 2023a). Different from convolutions, a ViT initially partitions the input image into patches, and processes these through self-attention and MLP blocks. ViTs have been observed to outperform convolutional networks at scale, despite arguably possessing less inductive bias (Dosovitskiy et al., 2020). This lack of inductive bias can be partially overcome through the introduction of "soft" inductive bias, which proves to be beneficial especially during the early phase of their training (d'Ascoli et al., 2021). Similarly to their counterparts in natural language processing, ViTs also exhibit predictable scaling behavior (Zhai et al., 2022; Dehghani et al., 2023a; Alabdulmohsin et al., 2023).

In our work, we are interested in having models equipped with adaptive "shape" parameters. We focus on the patch size used to process images, as well as the underlying model width as the adaptive variables. Training with varying patch sizes has been previously considered by Beyer et al. (2023), resulting in a model that is robust to the choice of patch size. It is also common practice to pretrain a ViT at a medium resolution and then subsequently finetuning it at a higher resolution while keeping the patch sized fixed (thus changing the number of patches) (Dosovitskiy et al., 2020; Zhai et al., 2022; Alabdulmohsin et al., 2023). Model (width) expansion under composable function-preserving operations has been a case of study for a long time in machine learning (Ash, 1989; Mitchell et al., 2023). The principal objective in this case is to accelerate training (Kaddour et al., 2023; Geiping & Goldstein, 2023). Such expansion operations have also been proposed for the

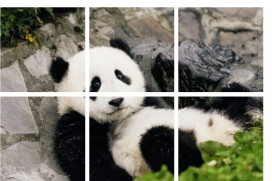 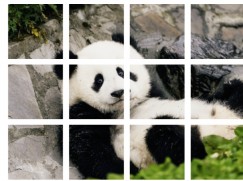 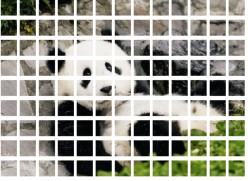 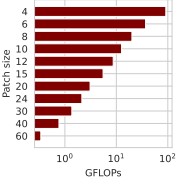

(a) Illustration of the effect of patch size on a given image.

(b) *ViT-B* FLOPs.

Figure 1: (Left) Patch sizes affect *how* ViTs process input images, while (right) having a prominent impact on the necessary compute of a forward pass.

Transformer architecture (Gesmundo & Maile, 2023; Chen et al., 2022) and have exhibited notable training speed-ups Gong et al. (2019); Yao et al. (2023); Wang et al. (2023); Lee et al. (2022); Shen et al. (2022); Li et al. (2022). Apart from determining *how* and *where* in the model this expansion should occur, a primary challenge is to resolve *when* to add new neurons. We advocate that an effective strategy for adjustments to the model shape should be informed by considerations of scaling laws and the performance gains achieved per additional unit of computational resources.

Orthogonal to our approach, various techniques have been proposed to accelerate both inference and training, particularly in the context of Transformer models. These methods encompass a spectrum of strategies, including weight quantization (Dettmers et al., 2022; Frantar et al., 2022) and pruning weights and context (Frantar & Alistarh, 2023; Anagnostidis et al., 2023) among others. Specifically for ViTs, Bolya et al. (2022) propose to merge tokens at different layers in the architecture and Dehghani et al. (2023b) propose to pack sequences of tokens together to optimize hardware utilization. Also d'Ascoli et al. (2021) propose to initialize ViTs differently, making them look more like convolutions. Other methods have also been proposed to beat scaling laws, including data pruning (Sorscher et al., 2022) or shaping models (depth vs width) more optimally (Alabdulmohsin et al., 2023). It is noteworthy that these approaches are supplementary to our methodology and can be effectively employed in conjunction to further enhance the efficiency of the training process. Discovery of optimal architectures have also been explored in the line of work of neural architecture search (Elsken et al., 2019). We want to highlight, however, that we are interested in more efficient training for a fixed architecture, the Transformer, that has established itself across different modalities.

## 3 VISION TRANSFORMER AND OPTIMAL PATCH SIZES

In this work, we focus on the family of Vision Transformers as they have become the de-facto dominant architecture for vision. ViTs process images $\mathbf{x} \in \mathbb{R}^{h \times w \times c}$, where $h, w$ are the height and width of the image in pixels and $c$ is the number of channels. Images are "patchified" into a sequence of $n$ tokens based on a specified patch size $p \in \mathbb{N}$, where $n = \lfloor w/p \rfloor \times \lfloor h/p \rfloor$, leading to a representation $\mathbf{x} \in \mathbb{R}^{n \times p^2 c}$. We illustrate the effect of different patch sizes in Fig. 1a. For simplicity, we only consider the case of equal height and width ($h = w$), although this constraint can be readily relaxed, as shown in Dehghani et al. (2023b). Each token is then linearly embedded with learnable parameters $\boldsymbol{W}_{emb} \in \mathbb{R}^{p^2 c \times d}$ where we refer to $d \in \mathbb{N}$ as the embedding dimension or width of the ViT. These embeddings are further enhanced with learnable positional encodings $\boldsymbol{W}_{pos} \in \mathbb{R}^{n \times d}$, enabling a ViT to learn the spatial structure of the tokens. The resulting embeddings are then processed by $L$ transformer blocks, consisting of a self-attention layer followed by an MLP that is shared across tokens. This specific structure of the architecture allows a ViT to generate predictions for token sequences of variable lengths, as is the case when dealing with images of different patch sizes.

**Fixed patch size training.** Different patch sizes come at different computational costs; the number of tokens $n$ scales with $\mathcal{O}(1/p^2)$ and thus processing inputs scales with $\mathcal{O}(1/p^4)$, due to quadratic

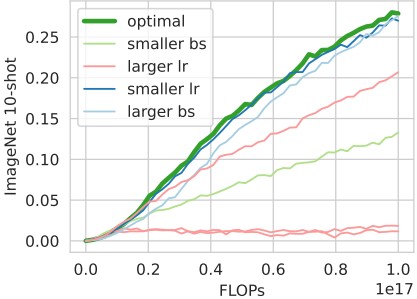

| Name | Width | Depth | Heads | Param (M) | GFLOPs | |
|---|---|---|---|---|---|---|
| | | | | | $X = 8$ | $X = 24$ |
| *V256-6/X* | 256 | 6 | 8 | 5.1 | 1.22 | 0.120 |
| *V192-12/X* | 192 | 12 | 3 | 5.6 | 1.43 | 0.136 |
| *V256-12/X* | 256 | 12 | 4 | 9.9 | 2.44 | 0.240 |
| *V384-12/X* | 384 | 12 | 6 | 21.8 | 5.25 | 0.538 |
| *V512-12/X* | 512 | 12 | 8 | 38.6 | 9.13 | 0.953 |
| *V640-12/X* | 640 | 12 | 10 | 60.0 | 14.1 | 1.49 |
| *V768-12/X* | 768 | 12 | 12 | 86.2 | 20.1 | 2.14 |

(a) We optimize batch size, learning rate, and weight decay for each model configuration by running a greed search, for a small compute budget. More details are presented in the Appendix.

(b) Details on the ViT models we are training. We use the standard *s*, *S*, *Ti*, *B* model sizes, as well as other intermediate model sizes. To simplify and unify notation, we adopt the naming convention *Vd-L/X* for a Vision Transformer of depth $L$ and embedding dimension $d$. Here $X$ refers to the patch size.

Figure 2: (Left) Hyperparameters are optimized across model classes. (Right) The ViT models used for this study.

dependence on the input sequence length of the self-attention operation[1]. Consequently, a reduction in the patch size results in a substantial increase in the computational requirements for a forward pass. We illustrate this increase numerically in Fig. 1b. Using smaller patch sizes on the other hand often yields enhanced model performance when paired with enough compute. To explore this trade-off, we pre-train variously-sized Vision Transformers (see Fig. 2(b) for a summary) on the public *ImageNet-21k* dataset (Ridnik et al., 2021) using different patch sizes that are *fixed* throughout training. For computational efficiency and to avoid being bottlenecked by data transferring, we resize images to $h = w = 120$[2]. This allows us to use a range of patch sizes $p \in \{120, 60, 30, 24, 20, 15, 12, 10, 8, 6, 4, 3, 2, 1\}$ that exactly divide the input resolution. We use FFCV (Leclerc et al., 2023) to load images efficiently. During training, we employ data augmentation techniques – random cropping and horizontal flips – and report 10-shot error (denoted as $E$) on *ImageNet-1k* (Deng et al., 2009), as upstream and downstream performance may not always be perfectly aligned (Tay et al., 2022; Zhai et al., 2022). Although doing multiple epochs over the same data has been shown to be suboptimal in cases for language modelling tasks (Xue et al., 2023; Muennighoff et al., 2023), augmentations, as employed in our study, allow conducting multiple epoch training without a noticeable decline in performance, at least for the data and compute scales (up to 10 EFLOPs) that we are analysing here (Zhai et al., 2022).

When calculating compute $C$, we exclude the computations associated with the "head" of the network that map the embedding dimension to the number of classes (Kaplan et al., 2020). Additionally, we adopt the approximation that the FLOPs required for the backward pass are approximately equivalent to twice the FLOPs incurred during the forward pass. Here, we are optimizing for FLOPs, which can be extended across different types of hardware accelerators. FLOPs in general exhibit a high degree of correlation with accelerator time (see e.g. Fig. 4 (right) in Alabdulmohsin et al. (2023)), given a fixed computational efficiency of the evaluated models (Stanić et al., 2023). In our study, we focus exclusively on Transformer models, which are very hardware-efficient (Dosovitskiy et al., 2020). More details regarding the experimental setup are provided in Appendix A.

For a fixed model size, we fit power laws in terms of compute (which is proportional to the number of examples seen in this case) for every patch size individually. The power law takes the form[3]:

---

[1]In reality, complexity is $\mathcal{O}(1/p^4 \times d + 1/p^2 \times d^2)$. For most choices of patch size, we have that $d > n$, and $\mathcal{O}(1/p^2 \times d^2)$ is the dominant term.

[2]We expect a small decrease in performance compared to reported numbers in the literature due to this decreased resolution. For our most compute-intensive models (ViT Base variant) we get – top-1 accuracy on *ImageNet-1k* – 79.2 % when fine-tuning and 77.2 %, when training a linear model on top of the extracted embeddings. Steiner et al. (2021) report 80.42% fine-tuning performance for a *ViT-B*/16 model on $224 \times 224$ images trained for 30 epochs on *ImageNet-21k*, which already surpasses our maximum compute budget.

[3]As aforementioned our models are bound by data rather than the number of parameters.

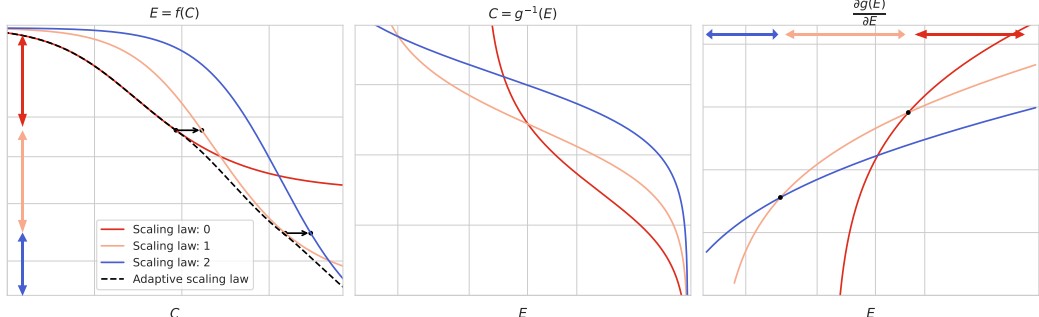

Figure 3: (Left) Different scaling law curves (function $f$ in Equation 1) corresponding to different training configurations. Arrows indicate points of transition between scaling laws. (Middle) We illustrate the inverse of the above function $g = f^{-1}$ for the same scaling law curves. (Right) We visualize the gradient of the inverse $\frac{\partial g(E)}{\partial E}$ for the same scaling laws. Taking the curve that maximizes the aforementioned gradient, leads to a partition of the space. Based on this partition, we can deduce a strategy on which scaling law to "follow" for each level of performance.

$$E_P = f_P(C) = a_P(C + d_P)^{-b_P} + c_P. \tag{1}$$

where the exponent $b_P$ dictates the speed of decay of the law and $c_P$ corresponds to the maximal reachable performance, i.e. when using infinite compute. We consider such a scaling law since we focus on varying solely a single shape parameter throughout training (here the patch size) while keeping the other dimensions fixed (e.g. the model size). After fitting the parameters $a_P, d_P, b_P, c_P > 0$, we can predict downstream performance $E_P$ (*ImageNet-1k* 10-shot top-1 unless otherwise stated) as a function of compute $C$ in FLOPs. We display the results for the *V640-12* model in Fig. 4 and Fig. 5 (we report the same plots for more model sizes in Appendix B). From the scaling laws, it is evident that different patch sizes are optimal at different levels of compute. Or in other words, *for a given performance level, different patch sizes yield different improvements for the same additional compute*. Given that insight, a very natural question emerges:

*Can we traverse between the scaling laws more efficiently by allowing for adaptive patch sizes?*

## 4 ADAPTIVE PATCH SIZES AND TRAVERSING SCALING LAWS

In this section, we will detail our strategy to efficiently traverse scaling laws. While we specialise the discussion here to the case of adapting the patch size (and leveraging the corresponding scaling law), the outlined strategy can in principle be extended to any change in "shape". Indeed, we will discuss the mechanism of growing a ViT in terms of width in Sec. 5.

**Adaptive Patch Size.** In order to allow for a smooth traversal of different laws, we first need a mechanism that enables mapping a ViT $f_P$ with patch size $P$ to a ViT $f_Q$ with patch size $Q$, while ideally not degrading performance, i.e. $f_P \approx f_Q$. *FlexiViT* introduced in Beyer et al. (2023) achieves precisely that. First, it is important to realise that solely the patch embedding $\boldsymbol{W}_{emb}$ and the positional encodings $\boldsymbol{W}_{pos}$ require adaptation. *FlexiViT* first defines both $\boldsymbol{W}_{emb}$ and $\boldsymbol{W}_{pos}$ for a fixed base patch size. In every forward pass, depending on the patch size, the base embedding parameters $\boldsymbol{W}_{emb}$ are resized based on the pseudo inverse of the resizing matrix. Similarly, the base positional encodings $\boldsymbol{W}_{pos}$ are bi-linearly interpolated, enabling the model to change the patch size without a strong performance degradation. We refer to Beyer et al. (2023) for more details.

**Traversing Scaling Laws.** Denote the family of scaling laws by $\{f_P\}$ with $P$ denoting the patch size. Each law maps a given level of compute $C$ to the predicted downstream performance $E_P = f_P(C)$. Consider the inverted laws $g_P(E) := f_P^{-1}(E)$, predicting for a given level of desired

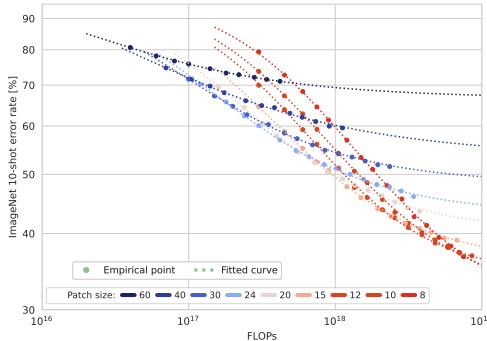 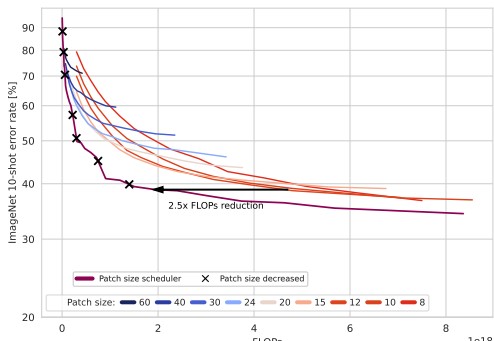

Figure 4: Downstream performance as a function of compute for the $V640\text{-}12$ model and different patch sizes. We use a log-log scale.

Figure 5: Downstream performance of the $V640\text{-}12$ trained with our patch size scheduler, and its potential benefits.

performance $E$, how much compute $C$ needs to be invested to achieve it. We aim to maximize the descent in performance $E$ at the current error level $E^*$. We thus compute the partial derivatives

$$q_P(E^*) := \left.\frac{\partial g_P(E)}{\partial E}\right|_{E=E^*} \quad \forall P. \tag{2}$$

Maximising $q_P$ over the patch size $P$ partitions the error space disjointly (we assume $E$ is the classification error taking values in $[0, 1]$),

$$[0, 1] := \bigcup_P E_P,$$

where $E_P \subset [0, 1]$ denotes the set where the patch size $P$ achieves the highest gradient. This partition naturally gives rise to a scheduler for the patch size, which empirically turns out to be monotonic (i.e. starting from the largest patch size for large classification error values and ending with the smallest for small classification errors), which is expected based on the observations in Fig. 4. We visualize the strategy in Fig. 3.

**Scheduled training.** We now test the devised strategy in a practical setting by pre-training variously-sized ViTs on *ImageNet-21k* using our patch size scheduler. We use the same training setup as for the fixed patch size experiments and let the scheduler consider patch sizes $P \in \{60, 40, 30, 24, 20, 15, 12, 10, 8\}$. We display *ImageNet-1k* 10-shot error rate as a function of compute $C$ for the model $V640\text{-}12$ in Fig. 5 (we provide results for all models in the Appendix B). The crosses denote the points where the scheduler switches patch size. We observe a significant improvement in terms of compute-efficiency, allowing for upto a 2.5 FLOPs reduction to achieve the same performance through training. While switching patch sizes might initially lead to a small degradation in performance due to changes of the entropy in the self-attention layers, in practice this deficit is very quickly overcome as the image is parsed in a more fine-grained manner. Such a degradation is thus not even visible in Fig. 5[4]. To facilitate a comparison across all model sizes at once, we further visualise the compute-optimal barrier for both fixed and scheduled training in Fig. 8. By compute-optimal, we refer to a model that optimally trades off model size, patch size and number of samples seen for a given level of compute $C$, i.e. achieving the lowest error $E$. We observe that the optimally scheduled models significantly outperform the optimal static models, halfing the required compute to optimally train a ViT-Base model (up to our budget of compute).

**Is the schedule optimal?** While our scheduler improves over the individual models, it is not clear yet that it does so in an optimal sense, i.e. can other schedules achieve similar benefits? Beyer et al. (2023) also employ a patch size scheduler but use a uniformly random sampling of the patch size

---

[4]Differences in the effective receptive field of each patch are typically mitigated by cropping as a component of the training procedure.

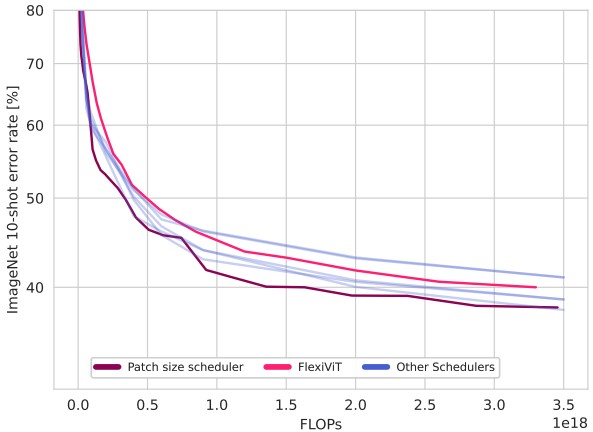

Figure 6: We compare performance as a function of training compute against the scheduler of *Flex-iViT* and other suboptimal schedulers. Irrespective of the current patch size in the scheduler, we use our smallest patch size (i.e. 8), when evaluating *FlexiViT*.

at every step. We compare against their model *FlexiViT* in Fig. 6. We observe that our scheduler indeed remains optimal, which is expected; *FlexiViT* targets a lower inference cost by making the model robust to many patch sizes (hence the random scheduler). Compute-optimality is not their objective. We further compare against a simple monotonic but linear as well as logarithmic patch size scheduler, i.e. for a given amount of total compute, we evenly (or logarithmically) space the transition points throughout training. This way we assess whether simply any monotonic scheduler leads to the same improvements, or whether the position of the transition points matter. We display the results in Fig. 6. We again observe that our scheduler remains optimal, carefully determining the transition points based on the scaling laws thus indeed leads to a significant improvement.

**Smaller Patch Sizes** Undeniably, the choice of patch size affects the inductive bias of ViTs (in general the level of tokenization in the input affects the inductive bias of any Transformer model). It controls the level of computing on the input and therefore the level of details we are interested in extracting from an image. The patch size also controls the overall sequence length $n$ processed by the Transformer model, and therefore the degree of weight sharing between the parameters. Our previous laws clearly show that smaller patch sizes lead to better performance in high-compute areas. But does this trend also extend to even smaller patch sizes? We explore this question empirically by using the same experimental setup and pre-training on even smaller patch sizes $P \in \{6, 4\}$ in addition to the previous results. We display the results in Fig. 7. We observe that while some absolute gains in performance can still be achieved with patch size 6, the additional required amount of compute is extremely high. For the even smaller patch size 4 one actually starts to lose in performance as can be seen from plotting the intercepts $C_P$ of the corresponding scaling laws. The behaviour of performance with respect to patch size is thus only monotonic up to a certain point, and performance may actually worsen beyond that. This is in contrast to other scaling parameters such as the number of samples or the model size that usually offer a monotonic behaviour in performance when scaled appropriately.

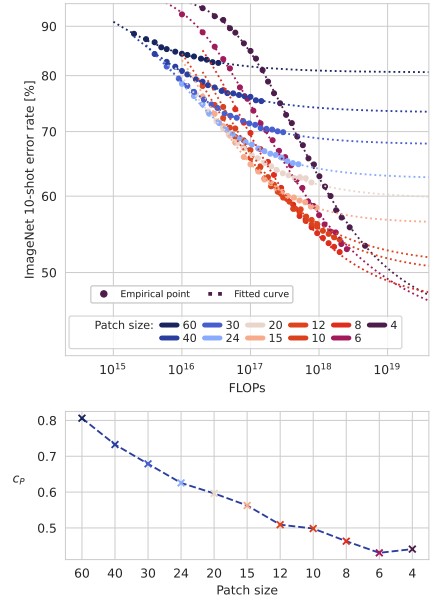

Figure 7: We train the *V256-6* with smaller patch sizes. This does not lead to a monotonically better performance.

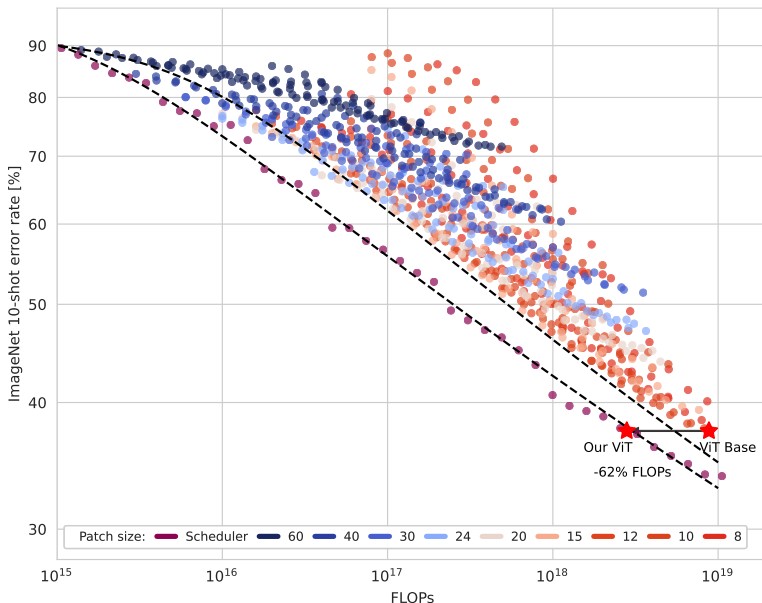

Figure 8: Compute-optimal static and scheduled models for various patch and model sizes. We plot using a log-log scale.

## 5 ADAPTING MODEL WIDTH

To further verify the efficiency of our approach, we study a different "shape" parameter of the Vision Transformer, the underlying width $d$ (or embedding size).

**Adapting width.** Similarly to the patch size, we need a mechanism that maps a transformer of smaller width $d_1$ to a transformer of larger width $d_2$. This is a very well-studied problem, especially in the case of natural language processing and many schemes have been proposed (Gesmundo & Maile, 2023; Chen et al., 2022; Gong et al., 2019; Yao et al., 2023; Wang et al., 2023; Lee et al., 2022; Shen et al., 2022; Li et al., 2022). Here, we focus on the simplest approach where we expand the initial model $d_1$ by adding randomly initialized weights (see Appendix B for details). This certainly does not preserve the function exactly and indeed we observe some drop in performance after adapting the model (see Fig. 10). On the other hand, we also notice that the expanded model quickly recovers, and hence conclude that while not ideal, this simple expansion mechanism suffices for our setting[5].

**Scaling width.** The role of the model width and its associated scaling properties are very well understood in the literature (Zhai et al., 2022; Alabdulmohsin et al., 2023). Nevertheless, we repeat the scaling study for our own experimental setup and pre-train Vision Transformers of various widths and training durations on *ImageNet-21k*. We use the same experimental setup as detailed in Sec. 3. In Fig. 9 we report 10-shot *ImageNet-1k* error as a function of compute for a fixed patch size $P = 20$. More details and results for different patch sizes are provided in the Appendix B. We again observe that different model widths are optimal for different levels of compute, similarly offering the potential for computational speed-ups by adapting the shape throughout training.

**Scheduling width.** It is worth noting that strategies for expanding models during training have been previously explored. However, the critical question of *when* this expansion should occur has largely remained an issue of controversy. Our approach then offers a straightforward and principled solution. We consider three width settings $d \in \{192, 384, 768\}$ and devise our scheduler based on the scaling law as outlined in Sec. 4. We display the obtained optimal schedule and the actual

---

[5]In practice we found that proposed function preserving schemes that mostly depend on zero-initializing weights in the network, e.g. (Gesmundo & Maile, 2023) perform suboptimally and do not allow the network to properly train.

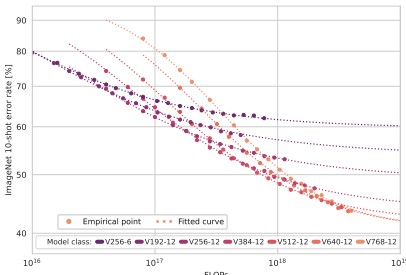
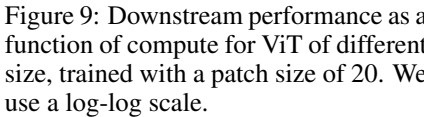

Figure 9: Downstream performance as a function of compute for ViT of different size, trained with a patch size of 20. We use a log-log scale.

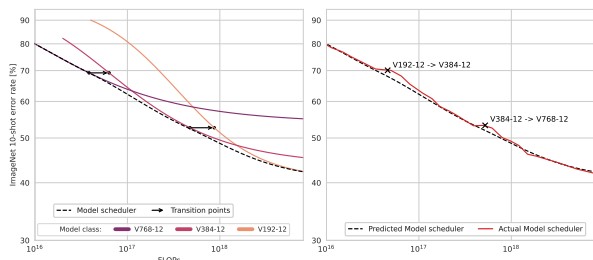

Figure 10: The theoretically predicted scheduled performance (left) and the actually obtained (right) performance. While transitions are less smooth, the model based on the scheduler quickly recovers back to the predicted law.

resulting performance in Fig. 10. As remarked previously, changing the model width does lead to a momentary deterioration of the performance, but smoothly recovers back to the predicted performance. We again observe that the scheduled model remains optimal throughout training when compared against the static models, but results are slightly less pronounced compared to the patch size scheduling.

## 6   CONCLUSION

In this work, we have explored strategies that leverage models with varying shape parameters as training progresses. By efficiently traversing neural scaling laws, we demonstrated how shape parameters such as patch size and model width can be optimally scheduled, leading to significant improvements in terms of the required level of compute. We further observe that such scheduled models perform compute-optimally compared to their static parts throughout training, further demonstrating that scheduling can get the best out of all the shape parameters. The proposed strategy is very flexible and applies to any shape parameter that admits a smooth mechanism to transform between two differently "shaped" models. We thus envision a wealth of potential future work applying our scheduling strategy to different shape parameters such as depth, sparsity or a combination of several parameters. We believe that such scheduling strategies are a timely contribution in light of the ever-growing demand of deep learning for more computational resources.

## 7   LIMITATIONS

We detail the limitations of our work to the best of our knowledge.

- We used greedy search with a small compute budget to get optimal hyper-parameters per model class. In practice, optimal parameters can change throughout training, e.g. in the literature it has been observed that larger batch sizes can be beneficial during late stages of training (Hoffmann et al., 2022; Zhai et al., 2023).

- In order to determine the optimal scheduler for a given shape parameter, knowledge of its scaling behaviour is needed, which comes at a high computational cost. On the other hand, the scaling behaviour of many shape parameters has already been established (e.g. width, depth, MLP-dimension (Alabdulmohsin et al., 2023)) and can readily be used in our scheduler.

- Accurately predicting compute-optimal models, requires one to accurately schedule the learning rate throughout training. As we are interested in low-budgets of computes we do not schedule the learning rate nor embark on a cooldown phase (Zhai et al., 2022), as this would constitute a large fraction of the overall training time. We expect learning rate schedulers may shift our conclusion but not the outcome and takeaway message.

- While we observe that the scheduled models are compute-optimal throughout all of training (especially for the patch size), we observe the largest gains earlier on throughout training. Indeed, we do not expect our scheduled models to reach better performance for an infinite amount of compute.

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

# A  EXPERIMENTAL SETUP

We provide more details on the basis on which the experiments were conducted.

## A.1  TRAINING DETAILS

| PARAMETER | VALUE |
|---|---|
| OPTIMIZER | ADAM |
| BETAS | $(0.9, 0.999)$ |
| LABEL SMOOTHING | 0.2 |
| WEIGHT-DECAY HEAD | 0.01 |
| WEIGHT-DECAY BODY | 0.01 |
| WARM-UP | 1000 STEPS |
| CLIP GRADIENTS' NORM | 1.0 |
| UNDERLYING PATCH-SIZE SHAPE | 12 |
| UNDERLYING POSEMB SHAPE | 8 |
| BATCH SIZE | 256 |

Table 1: Hyper-parameters during training. "Underlying patch-size" and "Underlying posemb shape" refer to the flexible modules when training under a flexible patch size scheduler.

In Table 1 we showcase hyper-parameters used when training on *ImageNet-21k*. We optimized each of the parameters for the different model classes by training for different configurations for a fixed, small amount of compute $4 \times 10^{17}$ FLOPs. Some examples of such hyper-parameter search are illustrated in Fig 11. All experiments were conducted using *bfloat16*.

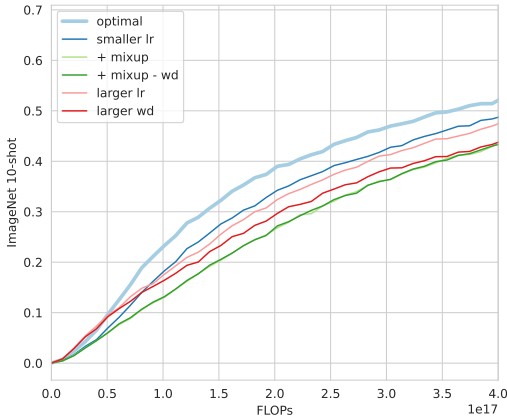

Figure 11: Hyper-parameter search for a fixed (and small) budget of compute.

## A.2  FINE-TUNING DETAILS

In Table 2 we showcase hyper-parameters used when finetuning on *ImageNet-1k*. For the few-shot results, we use the *linear_model.Ridge* function from *scikit-learn* with a regularization parameter of $1e^{-2}$.

## A.3  DATASET DESCRIPTION

We follow the protocol of Ridnik et al. (2021) to preprocess *ImageNet-21k*. It consists of roughly 12 million images and 11 thousand different classes. This is still considerably lower than the $\geq 29,593$ classes in the *JFT-3B* dataset. We experimented using different weight decay values for the body and the head, as proposed in Zhai et al. (2022) but found no significant difference. We attribute this

| PARAMETER | VALUE |
|---|---|
| OPTIMIZER | SGD |
| LEARNING RATE | 0.03 |
| MOMENTUM | 0.9 |
| WEIGHT DECAY | 0.0 |
| NUMBER OF STEPS | 20000 |
| CLIP GRADIENTS' NORM | 1.0 |
| SCHEDULER | COSINE |
| BATCH SIZE | 512 |

Table 2: Hyper-parameters during fine-tuning on *ImageNet-1k*.

to the lower number of classes in the dataset we are training in and the choice of label-smoothing (although *JFT-3B* is also weakly labelled).

### A.4 TRAINING VS TEST ACCURACY

To demonstrate that overfitting is not an issue during training, we present training vs accuracy results during training in Figure 12. We note that die to the aforementioned pre-processing, classes are more balanced than in the original (raw) *ImageNet-21k* dataset.

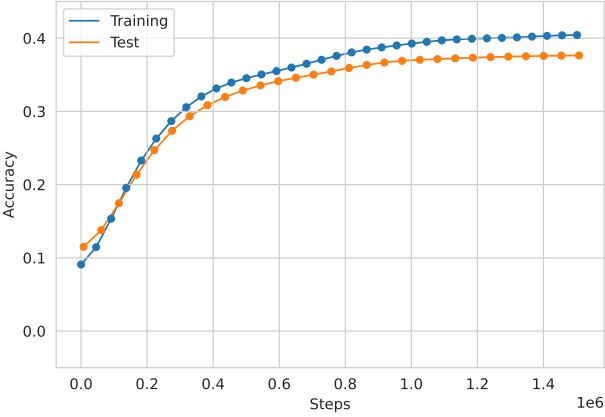

Figure 12: Training and test accuracy during training for the *V384-12* model using our patch size scheduler.

### A.5 SCALING LAWS

We fit functions of the form

$$E = a(C + d)^{-b} + c. \tag{3}$$

Similar to previous work, we resample points to be almost equidistant in the log-domain, in terms of FLOPs. We minimize different initialization using the *minimize* function in scipy (Virtanen et al., 2020), and choose the one that leads to the smallest error. The function to minimize is based on the *Huber loss* with $\delta = 1e^{-3}$.

### B ADDITIONAL EXPERIMENTS

**Patch size scheduler:** We present additional experiments on patch size schedulers in Fig. 13. For *FlexiViT*– similar to the original paper – we sample at every step a patch size from the set $\{8, 10, 12, 15, 20, 24\}$. We did not use smaller patch sizes due to computational constraints. Note that our patch size scheduler leads to significantly faster convergence across the model classes we

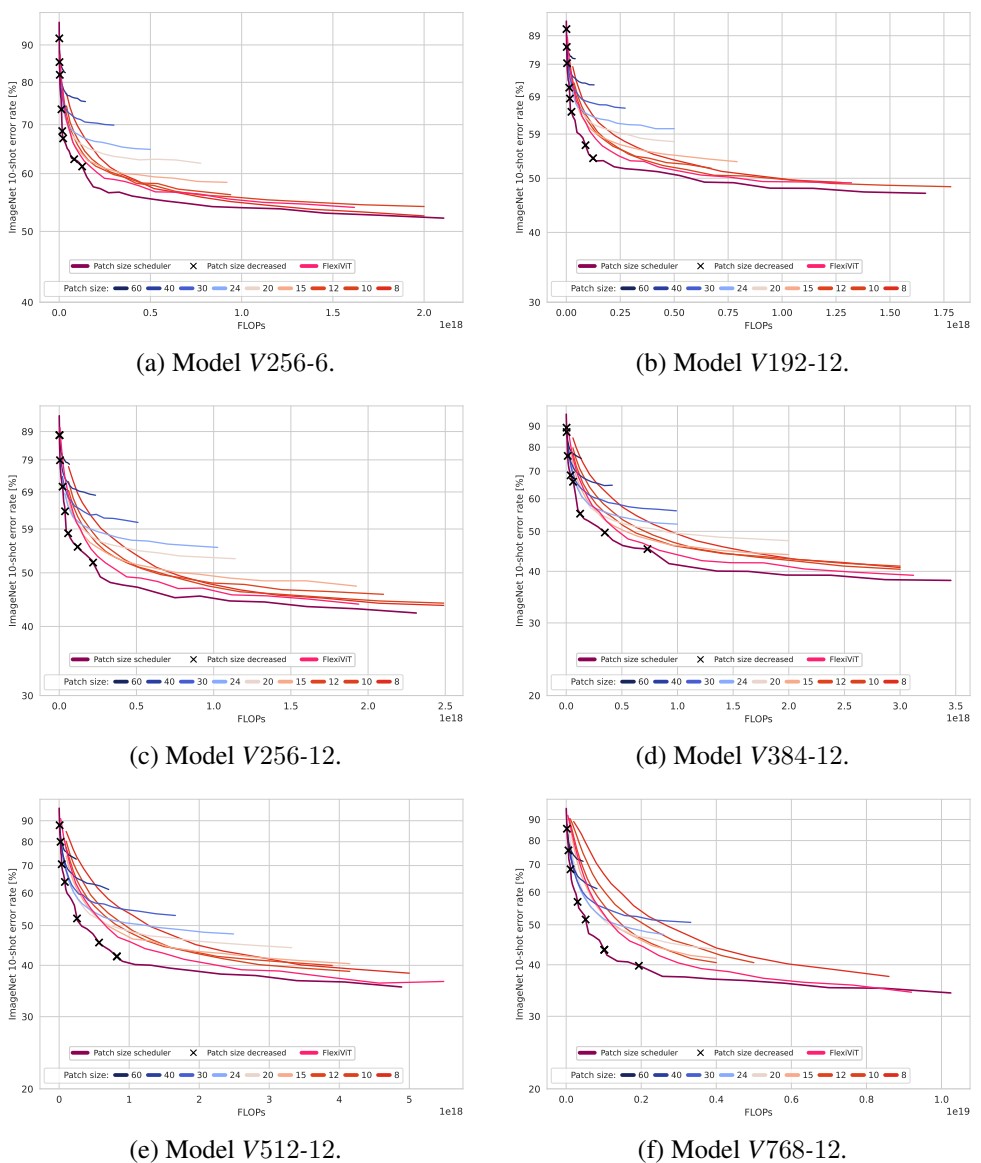

Figure 13: Patch size schedulers for all the remaining model classes analysed.

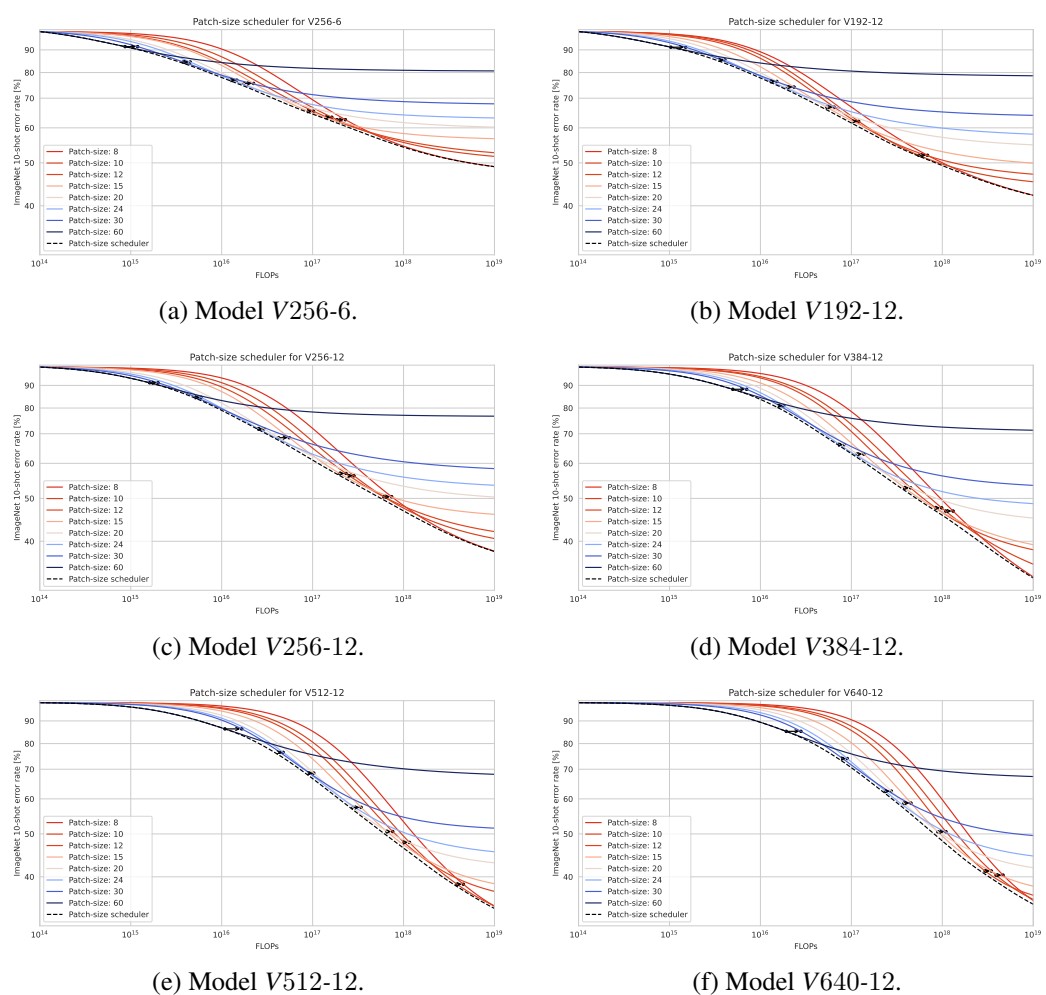

Figure 14: Fitted scaling laws and the predicted transition points that lead to the steepest descent.

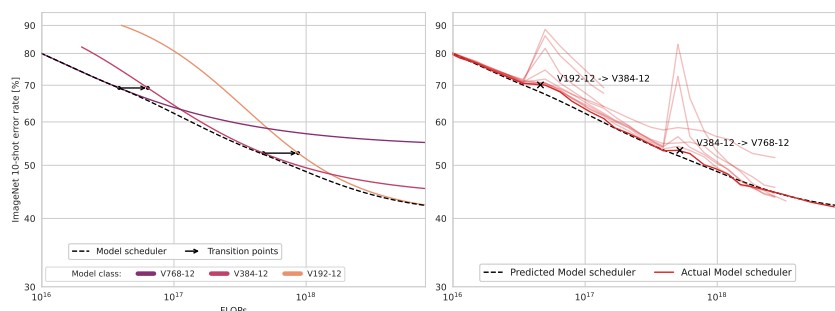

Figure 15: Different initialization schemes when expanding the width of the model. In practice, we set the variance of the new weights to be $\gamma\sigma^2$, where $\sigma^2$ is calculated from the pre-expanded weights $\boldsymbol{W}$, for different values of $\gamma \in \{0.25, 0.5, 0.75, 1, 1.25, 1.5\}$.

are analysing. We also present in Fig. 14, the fitted scaling curves and the points where changing the patch size leads to the steepest descent for different scaling laws.

**Model width scheduler:** Supplementary to the results in Section 5, we provide additional examples of width examples in Figure 16. Note that we do not touch on the (1) *where* to add the new weights and (2) *how* to initialize these new weights. Our approach simly defines a strategy on the *when* to expand the model and can be used in conjunction with any related works that provide answers to the previous (1) and (2) questions.

Regarding (1), we focus on models with constant depth (remember we are using the established *Ti*, *S*, and *B* sizes). Therefore, we do not add new layers but merely expand the weight matrices to the new embedding dimension. Our method is agnostic to *where* these weights are added, just on the final form of the scaling law. Note that there exist more optimal ways to expand the different components of a ViT model (Alabdulmohsin et al., 2023).

Regarding (2), there are numerous works on *how* to initialize the weights under a function preservation criterion. In our case, we found that zero-initializing weights, as commonly proposed, is significantly suboptimal. In practice, we expand the weights matrices by initializing the new entries in the weight matrices randomly based on the norm of the weights of the already learned weights. In more detail, linear layers are expanded as:

$$W' = \begin{pmatrix} W & W_1 \\ W_2 & W_{3,} \end{pmatrix}$$

where $W_1, W_2, W_3 \sim \mathcal{N}(\mu, \sigma^2 I)$, and $\sigma^2$ is calculated from $W$. This ensures better signal propagation in the network (He et al., 2015; Noci et al., 2022). The effect of this initialization can be important, but not detrimental, as illustrated in Fig. 15. When expanding the self-attention layers, we simply concatenate new heads, i.e. leave the heads that correspond to the previous embedding dimension unchanged. Again we stress that our method does not attempt to answer the question on *how* to initialize, and any method established in the literature can be used for this purpose.

## C    ADAPTING MULTIPLE MODEL SHAPE PARAMETERS CONCURRENTLY

We first present more results on which model configuration (number of parameters or patch size) leads to the most efficient training for different levels of performance in Figures 17 and 18.

Motivated by these insights, we ask the question: *Can we change both the model size and patch size during training, leading to even greater training compute savings?*

We present preliminary experiments here, and more specifically in Figure 19. We compare results when changing only the model width, only the patch size, or both the model width and patch size simultaneously. In every case we find the transition points, when the model shape should be adapted, using our proposed methodology. Changing both patch size and model width leads to the most significant improvements. For simplicity and clarity, we here consider model sizes in the set {*V*192-12, *V*256-12, *V*384-12} and patch sizes in the set {10, 20, 30, 40}.

We note that our method does not take into account momentary performance boost, when reducing the patch size and momentary performance deterioration when changing the model size, due to reasons highlighted in the main text. This justifies why changing only patch size can be better in some cases for the short term. As more compute is invested into the new model shape, these changes are counteracted.

## D    MORE SHAPE PARAMETERS

In this paper, when changing the model itself in Section 5, we adjusted just the width of the models. This was done to interpolate between existing in the literature model sizes (Dosovitskiy et al., 2020). We note however that our proposed technique is general and can be applied to any "shape" parameter of the model or any optimization choice. In Figure 20, we present preliminary experiments on depth as another such "shape" parameter. More specifically, we add new layers in the Transformer at specified points in training, as defined by our method. In this case, we can easily do this in

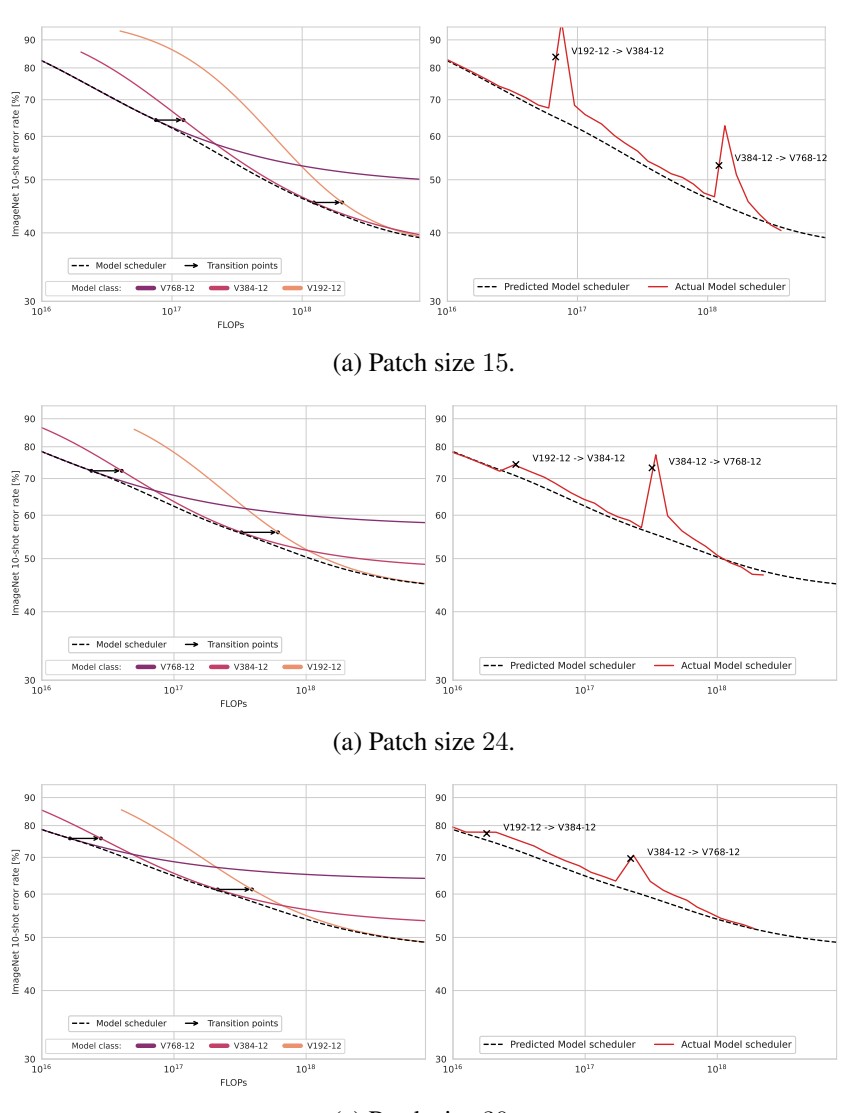

(a) Patch size 15.

(a) Patch size 24.

(c) Patch size 30.

Figure 16: Width scheduler for models trained with different patch sizes. We expand the model width twice, as done in Section 5. The transition points of the expansion are based on our maximum descent rule.

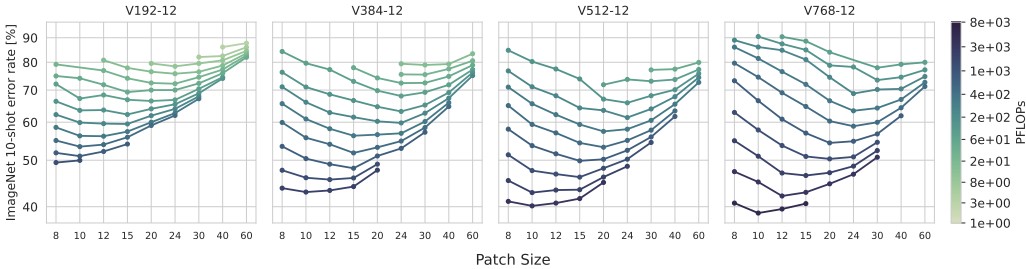

Figure 17: IsoFLOPs curves for different size ViTs trained with different constant patch sizes. Note how larger patch sizes are favoured for smaller total FLOPs, while smaller patch sizes become more efficient as total FLOPs increase. Larger model sizes also become more favourable as total FLOPs increase.

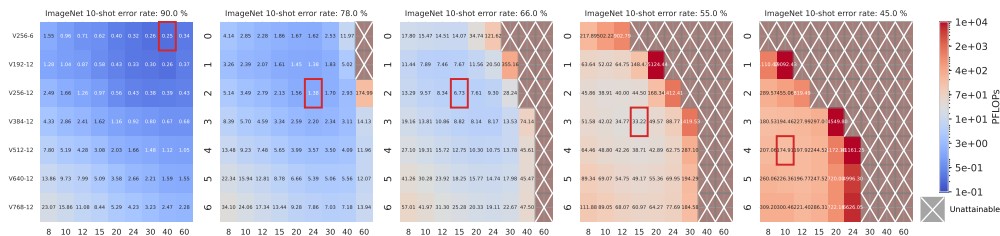

Figure 18: Values for $-\frac{\partial g_P(E)}{\partial E}$ in Equation 2. Values indicate how many FLOPs are required for a proportionate increase in performance (i.e. drop in error rate).

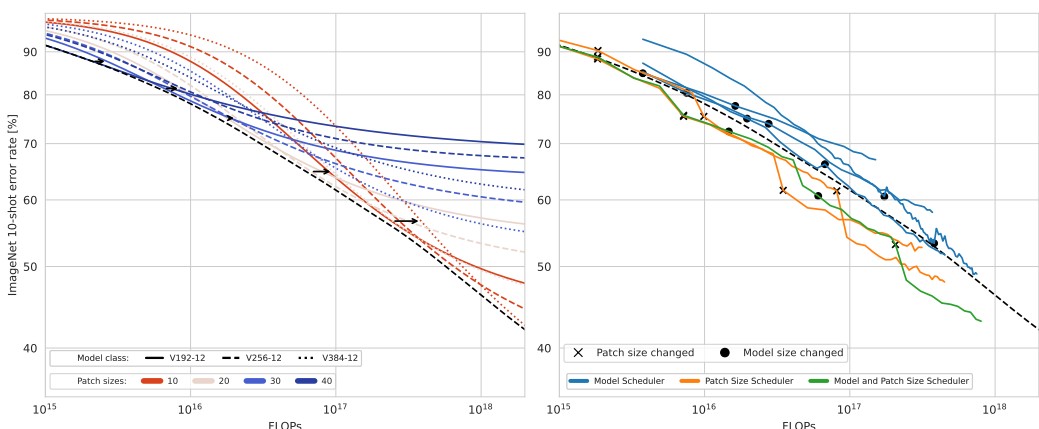

Figure 19: Changing both model width and patch size during training further accelerates training.

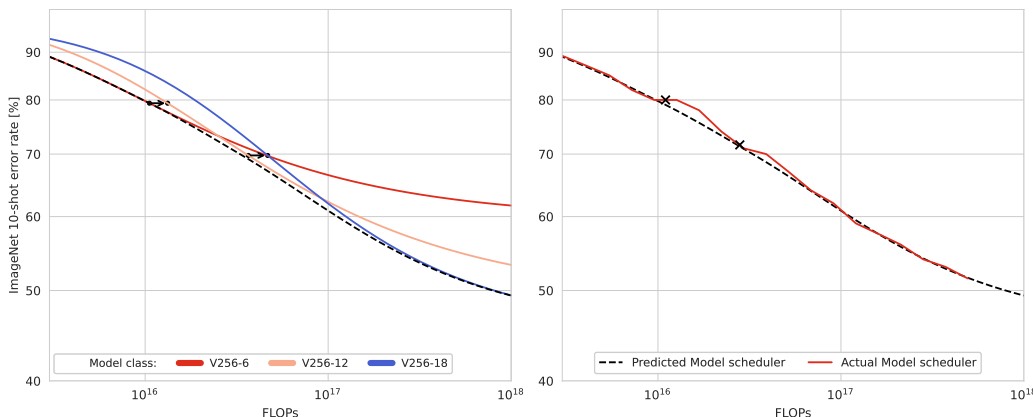

Figure 20: Traversing scaling laws for model of varying depth. X's here denote adding more layers to a model.

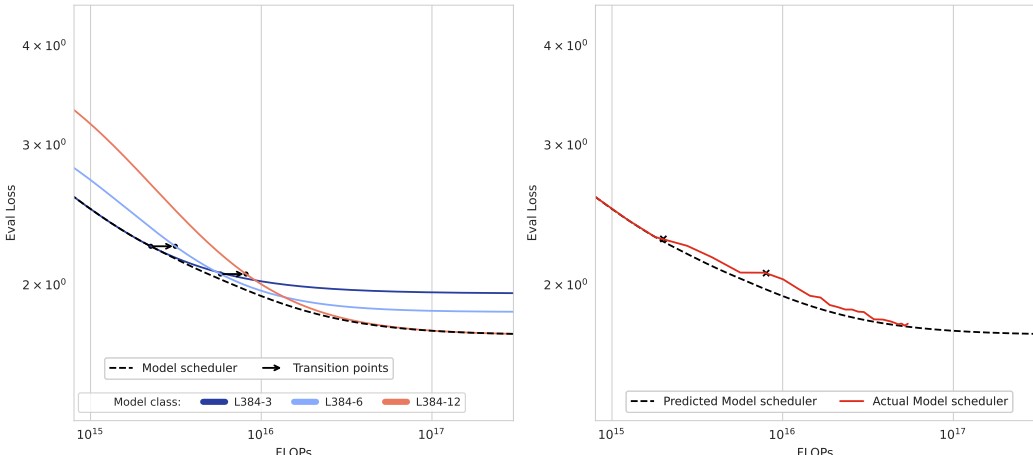

Figure 21: Traversing scaling laws for a language modelling task. Here we use the $LW$-$D$ naming convention to denote a language model with width $W$ and depth $D$. X's denote at what point the model's depth was increased.

a functional-preserving way, initializing the new layers randomly and adding scalar values in the Attention and MLP blocks initialized as ReZero (Bachlechner et al., 2021). We note that expanding the width versus the depth of a Vision Transformer is not part of the scope of this paper. Existing literature (Alabdulmohsin et al., 2023) addresses such issues.

# E  BEYOND VISION TRANSFORMERS

In this paper, we focused on Vision Transformers, as the patch size offers another "shape" parameter that can be easily adapted. We note, however, that our proposed technique is general and applicable across modalities and different shape parameters. Here, we present preliminary results on how the model width can be modified during training, as done in Section 5, for a language modelling task.

We adopt the experimental setup of He & Hofmann (2023) and train decoder-only Transformer models of width $384$ and different depths $3, 6, 12$. Then, we extract our optimal scheduling rule and train a model that dynamically adjusts its width during training. Results are presented in Figure 21 and again demonstrate how such a technique can be applied outside the field of vision.

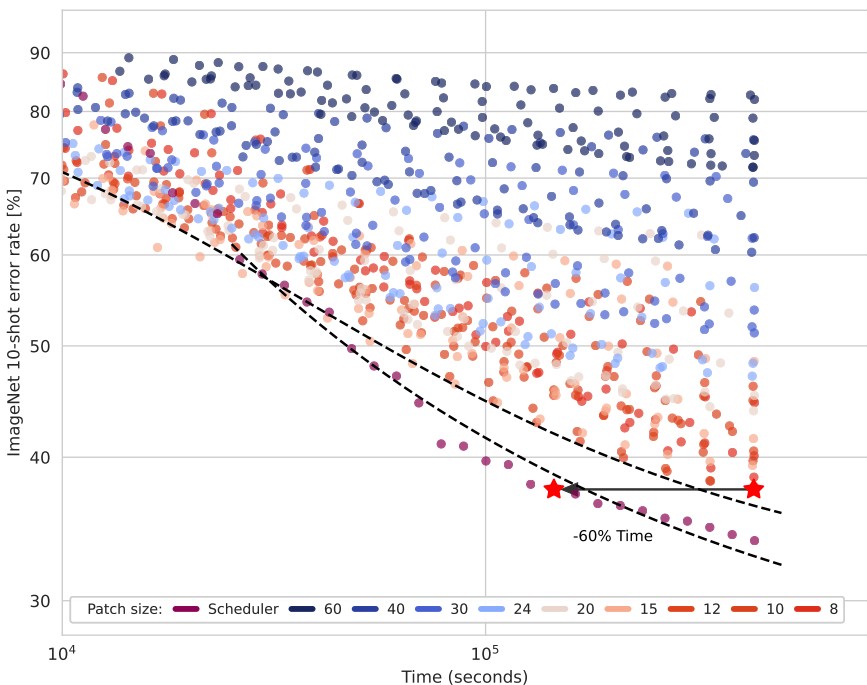

Figure 22: Same plot as 8 but with time instead of FLOPs in the x-axis.

## F ENVIRONMENTAL IMPACT

In order to estimate the carbon footprint of our training, we follow the recipe detailed in Touvron et al. (2023). Specifically, we approximate the Watt-hours (Wh) used as

$$Wh = GPU\text{-hours} \times GPU\text{-power-consumption} \times PUE$$

where PUE refers to Power Usage Effectiveness. Following Touvron et al. (2023) we set this quantity to 1.1. In order to enable comparisons across different works, we use the national US average carbon intensity factor of $0.385\ kg\ CO_2eq/KWh$ and we thus estimate the amounts of carbon emissions as

$$tCO_2eq = MWh \times 0.385.$$

We compare our adaptively trained model against standard training of the compute-optimal model, in this case, the ViT Base model with patch size 8. The model requires $\approx 120$ GPU-hours with an average consumption of $\approx 280W$ with the default training. Our adaptive training requires roughly $40\%$ of GPU-hours, i.e. $\approx 48$ GPU-hours while enjoying the same average consumption $\approx 280W$. This leads to $\approx 0.036MWh$ for ViT-Base and $\approx 0.014MWh$ for our adaptive training. Thus, the default training of the ViT Base model causes carbon emissions of $0.014tCO_2eq$ and our training $0.006tCO_2eq$.

We also added a figure for Time in Fig. 22. We note that time and FLOPs are usually highly correlated (Alabdulmohsin et al., 2023). This relationship also depends on the type fo hardware and the mode it is operating in, i.e. whether we are memory-bound, whether data loading is the bottleneck etc.

