# OpenReview forum: "Navigating Scaling Laws: Accelerating Vision Transformer's Training via Adaptive Strategies"
_ICLR.cc/2024/Conference — Submitted to ICLR 2024_

### Official Review · Reviewer_oGK8 · 2023-10-22

**Soundness:** 3 good
**Presentation:** 4 excellent
**Contribution:** 2 fair
**Rating:** 3
**Confidence:** 3

**Summary:**

The main argument of the paper is that, by being guided by scaling laws, one can develop compute-optimal adaptive models that surpass the performance of their static counterparts. The paper then proposes an "adaptive" ViT model that can alter its patch size during training. The authors claim the model optimally navigates the underlying scaling laws, thereby reducing the computational resources required to attain a specified performance level.

**Strengths:**

S1. Relevance: With the increasing computational demands of state-of-the-art models, finding ways to optimize compute resources without compromising performance is very important. This work directly addresses the issue, with a somewhat unique take on the problem.

S2. Theoretical motivation: The paper's emphasis on neural scaling laws as the guiding principle behind adaptive modeling provides a good basis and motivation for the proposed approach. This grounding might inspire new ideas in this direction.

S3. Empirical results: The proposed adaptive model seems to surpass its static counterpart for most patch sizes which is a promising result. This could make the method useful for practitioners.

**Weaknesses:**

W1. Novelty: The concept of adaptive models that can modify their shape during training seems very similar to Neural Architecure Search (NAS) which isn't covered at all in the related work or in the comparisons. See [1] for a comprehensive overview of NAS.

W2. Complexity: While adaptive models sound promising, they might introduce additional complexities in terms of training dynamics and hyperparameter tuning. I appreciate that the authors have mentioned some of these issues in the "limitations" section, but I feel a more comprehensive treatment is necessary especially related to how the optimal scheduling will change for other tasks / architectures.

W3. Overfitting concerns: Altering a model's shape during training could lead to overfitting issues. I think the paper should address this concern more explicitly eg. what do the train / test gaps look like for the adaptive model.

[1] Elsken, Thomas, Jan Hendrik Metzen, and Frank Hutter. "Neural architecture search: A survey." The Journal of Machine Learning Research 20.1 (2019): 1997-2017.

**Questions:**

Questions and recommendations:

1. How do NAS approaches relate to neural scaling laws and the method proposed in the paper?
2. At least some comparison to state of the art NAS methods is needed, as this would strengthen the paper's claims and make its contributions clearer.
3. The authors should provide a more in-depth overview of the complexities introduced by adaptive models, potentially with guidelines to address these.
4. Is overfitting an issue? What does the train / test gap look like for the models?

---

> ### Author Response · Authors · 2023-11-17
>
> We thank the reviewer for the concerns raised.
>
> 1. **Comparing to NAS:** We thank the reviewer for pointing out this connection. We believe however that there is a misunderstanding and we apologise for the confusion in advance. Compute-optimality, as considered in our work, as well as in the relevant literature [1, 2, 3, 4, 5, 6], concerns finding a model **within** a specified family of models (e.g. the family of Vision Transformers), which has optimal test error for a pre-specified level of **compute** (e.g. number of FLOPs/training time etc.). We are not concerned with finding **new** optimal models but rather with how we can **reach them more efficiently**, i.e. by training with **fewer** FLOPs.
> Neural architecture search on the other hand is primarily concerned with finding a new optimal model architecture for a given task by changing architectural components such as the activation function, normalisation layers, the computational graph etc. Such an approach always focuses on finding a **single, fixed model** while we simply propose to reach such an optimal model through **smarter training**. We have expanded the related works section and made this point clearer.
>
> 2. **Additional complexity:** We agree that our method introduces more complexity since it involves training a series of models. We remark however that we **did not tune hyper-parameters** for the different models but used the **same set throughout training** (e.g. no change in learning rate, momentum etc.). One could argue that the transitioning times enhance complexity, but those are determined by the scaling law, which is precisely why we think our framework offers a powerful toolbox for model training. We observe our approach to further be stable across different modalities (see point 3 in global response), thus we believe that the introduced additional complexity is minimal.
>
> 3. **More overfitting:** Thank you for this question. We actually believe that the opposite is true. (1) When starting to train adaptively, the initial models are usually the lower capacity ones, e.g. we start with smaller width, smaller depth (see 2 in global response) and larger patch size, thus leading to less (potential overfitting). (2) Changing the model shape acts as a regulariser, as transitioning momentarily decreases test and training accuracy as a portion of the network is randomly initialised (at least for width and depth adaptation). (3) We are pre-training on a large corpus of images (around 15 million images) with data augmentation, thus overfitting in general is less of an issue in such a setting.
> We further back these arguments with empirical evidence in Figure 12, where we plot (upstream) training and test accuracies against training steps. We can indeed see that overfitting is not an issue with our approach.
>
> We hope that we could address the questions of the reviewer and are happy to further clarify remaining concerns. If not, we would be grateful if the reviewer would consider raising the score.
>
> [1] Kaplan et al., Scaling Laws for Neural Language Models
>
> [2] Hofmann et al., Training Compute-Optimal Language Models
>
> [3] Zhai et al., Scaling Vision Transformer
>
> [4] Alabdulmohsin et al., Getting ViT into Shape
>
> [5] Frantar et al., Scaling Laws for Sparsely-Connected Foundation Models
>
> [6] Smith et al., ConvNets Match Vision Transformers at Scale

---

### Official Review · Reviewer_oxcs · 2023-11-01

**Soundness:** 3 good
**Presentation:** 4 excellent
**Contribution:** 3 good
**Rating:** 6
**Confidence:** 4

**Summary:**

- The work is motivated by the following observation that the shape of the model remains fixed throughout the training process. Shape in this context refers to hyperparameters of the model that do not degrade performance when changed (such as width, depth and patch size).
- In turn, the paper proposes an adaptive training methodology to achieve equivalent performance for a specific model while using fewer computational resources (termed as FLOPs).
- The algorithm exploits the fact that the scaling laws proposed till now are in the form of a generalized power law. Thus, they compute the inverse function (as the fitted scaling law is bijective) and then compute the set where the shape parameter being tuned achieves the highest gradient. This set provides a scheduler for the shape parameter being tuned across training time.
- The paper leverages existing flexible transformer architectures like FlexVIT to tune shape-parameters like patch size and model width.

**Strengths:**

- Good motivation to the paper, the method makes intuitive sense to me. The evaluation performed is adequate.
- Excellent results. It appears that substantial improvements in FLOPs is observed compared to the baseline schedulers (fixed) and the FlexVIT scheduling method (which is the architecture they employ).
- The presentation of the work is very good. Visualizations are helpful in understanding the work.

**Weaknesses:**

- While the proposed method is interesting and makes intuitive sense, don't such shape parameters exist for transformer models in other modalities. An experiment to show that the scheduler generalizes to text or audio or other modalities would make this work far stronger.
- The work explores the axis being tuned one at a time (patch size or model width). It would be nice to show that the result holds when two shape parameters are scheduled together. Do the authors expect the trend to hold?
- While FLOPs are one measure of computational cost, they do not characterize all aspects of performance [1] (compared to training time, carbon emissions etc). In the context of this work, as the architecture is kept constant, I expect the assumption to hold (FLOPS are a proxy for computational cost), however, it would be good to have further discussion around this point.
- The work does not provide a discussion/results wrt potentially improved carbon emissions estimates and the overall end-to-end cost of their training procedure (e.g. accounting for their greedy search to get optimal hyper-parameters compared to other baseline methods). Showing just FLOPs presents only one perspective of computational cost.

[1] https://arxiv.org/abs/2210.06640

**Questions:**

Please clarify the questions and comments listed in Weaknesses.

---

> ### Author Response · Authors · 2023-11-17
>
> We thank you for acknowledging our experimental results and the very helpful feedback. We are excited to further improve our work based on your recommendations.
>
> 1. **More modalities:** Thank you for your comment, which prompted us to verify our framework further on text-based tasks. For a detailed description, we point to the global response to all reviewers, especially point 3 where we address it as well as the new section Appendix E in our work.
>
> 2. **Simultaneous shape adaptation:** This is a very good point! We verified that our framework can indeed handle such simultaneous adaptions by varying both the patch size and the model width. For more details, we point the reviewer to the global response to all reviewers, especially point 1.
>
> 3. **Measure cost not just through FLOPs:**  Thank you for pointing this out. We absolutely agree that fewer FLOPs do not necessarily imply shorter training time. As you correctly point out, we use the same underlying family of functions when we adapt the “shape”, so in this case, we expect a strong correlation between FLOPs and training time. For completeness, we display the corresponding relationship in Figure 22 where performance is plotted against compute measured in time (seconds) instead of FLOPs. As expected we observe the same qualitative picture. We will highlight this observation in the main text.
>
> 4. **End-to-end training cost:** Thank you for this suggestion. We have estimated the carbon emissions caused by (1) training a ViT-Base model in the standard manner and (2) training it with our adaptive technique. We find that emissions are reduced roughly by 60%. We provide a detailed derivation of this estimate in Appendix F.

---

### Official Review · Reviewer_76wg · 2023-11-05

**Soundness:** 2 fair
**Presentation:** 2 fair
**Contribution:** 2 fair
**Rating:** 3
**Confidence:** 3

**Summary:**

In this paper, authors introduce a new approach to training deep learning models that we call “adaptive training.” Adaptive training allows the model to change its shape during the course of training, which can lead to significant reductions in the required compute to reach a given target performance.

This work focuses on vision tasks and the family of Vision Transformers (ViTs), where the patch size as well as the width naturally serve as adaptive shape parameters. Authors demonstrate that, guided by scaling laws, one can design compute-optimal adaptive models that outperform their “static” counterparts.

They then propose a simple and effective strategy to traverse scaling laws, opting for the one that leads to the fastest descent, i.e. maximum performance gain for the same amount of compute. The work showcases the efficiency of our approach by optimally scheduling the patch size of a Vision Transformer, leading to significant reductions in the required amount of compute to reach optimal performance. They further confirm the validity of the approach by optimally scheduling the model width of a Vision Transformer.

The work demonstrates that adaptive training is a promising new approach to training deep learning models that can significantly reduce the computational resources required to achieve state-of-the-art performance.

**Strengths:**

The paper is based on a sound reasoning that having a static architecture while studying scaling laws might not lead to optimal use of compute. This paper proposes an alternative to consider architecture to be elastic which can be modified based on the insights from intermediate scaling laws.

**Weaknesses:**

Considering patch size and width of the transformer blocks are reasonable first choices to tune flops per training example. It would be interesting to consider other choices and their effects.

**Questions:**

I would like to see more ablations on other choices of the transformer architecture on their effect on optimal architecture.

---

> ### Author Response · Authors · 2023-11-17
>
> We thank the reviewer for acknowledging the simplicity and effectiveness of our approach.
>
> **More shape parameters:** We agree that using more “shape” parameters in our study would strengthen the results. To this end, we performed experiments on adapting the depth of the Vision Transformer. For more details, we point the reviewer to the global response to all reviewers, especially point 2.
>
> We hope that we could address the questions of the reviewer and are happy to further clarify remaining concerns. If not, we would be grateful if the reviewer would consider raising the score.

---

### Official Review · Reviewer_SgbS · 2023-11-07

**Soundness:** 2 fair
**Presentation:** 2 fair
**Contribution:** 2 fair
**Rating:** 3
**Confidence:** 3

**Summary:**

The paper proposes a “compute-optimal” model, where the characteristics of the network, such as width, depth, and patch size, could be changed/adapted during training. The authors argue that such an “adaptive” model can utilize the scaling property of the neural network, and achieve “optimal” computation status---in a way, it balances the tradeoff between the computation and the performance. The modification of the network mainly focus on ViT, and the authors schedule the width and patch size to reduce training computation resource. Specifically, for both patch size and the network width, the authors apply adapting, scaling, and scheduling strategies. Experiments on ViT of different patch sizes/widths show promising computation reduction.

**Strengths:**

- The paper focuses on an interesting topic in transformer network (ViT) training---how to tune the parameters---which are image patch size and network width---to achieve "optimal" training with "optimal" training resources.

- The study of previous work is new and very related to the topic.

**Weaknesses:**

- The authors have mentioned 3 contributions in the paper. However, these contributions could be summarized as one point---an adaptive strategy to change the patch size and the network width to adapt training.

- Many arguments in the paper lack theoretical connections, and only some empirical results were shown in the paper. I think if the paper focuses on empirical results, more experiments such as different network property changes, and different changes with multiple kinds of properties should be discussed and compared. If the authors keep current experiments, more applications or tasks could be included to further validate the proposed method.

- Following the above discussion, one example is: when adapting patch size, there are no width changes, and when adapting network width, there are no patch size changes. I wondered if the authors could do further experiments on various changes including cross adapting to make the argument stronger.

- Some of the limitations discussed in the paper could actually incorporated in this paper.

**Questions:**

There are also some ambiguities in the figures/paragraphs. For example, in Figure 3, what do those arrows mean?

My main concern is discussed above. Please follow this discussion to further improve the paper.

---

> ### Author Response · Authors · 2023-11-17
>
> We thank the reviewer for the points raised.
>
> 1. **Summarise contributions:** We would like to highlight that the devised framework applies to any “shape” parameter and not just width or patch size. We thus do believe that this is a contribution of its own due to its generality. We then verify our framework by specialising it to adaptive patch size as well as width. We agree that the latter two points can also be summarised as one. We have re-written the main text accordingly.
>
> 2. **More empirical evidence:** We agree that more experimental results would strengthen our work. To address this point, we have performed more experiments on (1) adapting the depth of the network, (2) adapting patch size and width simultaneously, as well as (3) adapting depth on text-based tasks. For more details, we point the reviewer to the global response to all reviewers.
>
> 3. **Simultaneous shape adaptation:** This is a very nice suggestion. We verified that our framework can indeed handle such simultaneous adaptions by varying both the patch size and the model width. For more details, we point the reviewer to the global response to all reviewers, especially point 1.
>
> 4. **Arrows in Figure 3:** We apologise for the confusion. For the left-most and right-most figure, the colorful arrows denote which partial derivative is maximal for a given level of error E. The black arrows on the other hand correspond to transition points where the shape of the model is changed. This figure purely serves as an illustration of our framework. We have updated the caption in the main text accordingly to clarify this.
>
> We hope that we addressed the questions of the reviewer and are happy to further clarify remaining concerns. If not, we would be grateful if the reviewer would consider raising the score.

---

### Author Response · Authors · 2023-11-17
**General Response**

We thank the reviewers for the valuable feedback that we truly believe has helped us improve our work. We have updated the paper, including more experiments that we hope further bolster the claims made throughout our work. We refer the reviewers to the newly created sections C, D, E, and F in the appendix.

1. **Simultaneous shape adaptation:** We thank the reviewers for this very nice suggestion! Our framework can indeed handle multiple shape parameters at once. We verified this by adapting the patch size and the model width of a Vision Transformer simultaneously and indeed observed that such a strategy leads to even more computational savings, leading to a reduction in FLOPs of 65% (Appendix C).  We refer the reviewers to Figure 19.

2. **More shape parameters:** In order to demonstrate the validity of our framework further, we included the depth of the network as a “shape” parameter in our experimental study (Appendix D). We use the same experimental setup as for patch size and model width and grow the network over the course of training from a starting depth of 6 to the final depth 18, using the set of values {6, 12, 18} as allowed depths during training. We show in Figure 20 how performance matches the predicted scaling.
We would like to highlight further that our method works even more generally, allowing one to switch between different training objectives (e.g. self-supervised to supervised) or even datasets (pre-training to target) or adjusting hyperparameters such as batch size during training. We leave such explorations to future work.

3. **More modalities:** We agree with the reviewers that verifying our framework on more modalities would strengthen the results. As a step towards this, we also evaluated our shape-adaptive strategy on text-based data when growing the model’s depth (Appendix E). We perform auto-regressive learning on the “Code Parrot” dataset, following the evaluation protocol of [1]. We use upstream evaluation loss as the metric of interest. We display the corresponding results in Figure 21. Varying context size (the natural “dual” to patch size in vision) or varying the context window of the attention operations are also interesting directions that we leave for future work.

If no further concerns remain, we kindly ask the reviewers to reconsider their scores.

[1] He, Bobby, and Thomas Hofmann. "Simplifying Transformer Blocks." arXiv preprint arXiv:2311.01906 (2023).

---

### Author Response · Authors · 2023-11-21

The discussion period is ending soon, and we still haven’t heard back from any of the reviewers. The reviewers provided questions which we addressed in our rebuttal, and it would be great to have a discussion in case some points have remained unclear. We would really appreciate it if you provide feedback at your earliest convenience.

Best wishes,
The authors

---

### Meta-Review · Area_Chair_JSie · 2023-12-11

**Metareview:**

In this paper, the authors introduce a new training of deep neural networks that relies on adaptively changing the shape and size of the model during the course of training. The goal of the training algorithm is to trade off compute versus predictive performance during the course of training. The authors demonstrate the efficacy of their results by training a series of vision transformer models for image classification tasks. The reviewers commented positively on the question being addressed by the authors and the underlying hypothesis of the work. The reviewers also commented negatively on the limited scope of the experiments in terms of identifying more shape parameters that may be adapted during training, as well as concerns about the complexity of the method, and concerns about overfitting. The authors' rebuttals were not responded to by the authors, so the AC examined the concerns raised by the reviewers to see if this would pass the bar of acceptance. At face value, the AC agrees with the reviewers that additional experiments are needed to justify the complexity of this method and make for stronger empirical results. One item of concern is how the authors could provide evidence that their method generalized beyond the narrow parameters of vision transformers and image classification. Given the uniformly low scores and the review by the AC, this paper does not warrant acceptance by the conference. The authors however are encouraged to incorporate their follow up experiments and do their best to address some of these concerns and resubmit to a future workshop or conference.

**Justification For Why Not Higher Score:**

Unconvincing experiments

**Justification For Why Not Lower Score:**

N/A

---

### Decision · Program_Chairs · 2024-01-16

Reject